# In Situ Formation of Injectable Gelatin Methacryloyl (GelMA) Hydrogels for Effective Intraocular Delivery of Triamcinolone Acetonide

**DOI:** 10.3390/ijms24054957

**Published:** 2023-03-04

**Authors:** Chaolan Shen, Xuan Zhao, Zewen Ren, Bing Yang, Xiaohui Wang, Andina Hu, Jie Hu

**Affiliations:** State Key Laboratory of Ophthalmology, Guangdong Provincial Key Laboratory of Ophthalmology and Visual Science, Zhongshan Ophthalmic Center, Sun Yat-sen University, Guangzhou 510060, China

**Keywords:** gelatin methacryloyl, triamcinolone acetonide, retina hydrogel system, hydrogel drug release, intraocular drug delivery, injectable gel

## Abstract

A novel drug delivery system designed for intraocular injection, gelatin methacryloyl (GelMA), has attracted much attention due to its sustained-release character and low cytotoxicity. We aimed to explore the sustained drug effect of GelMA hydrogels coupled with triamcinolone acetonide (TA) after injection into the vitreous cavity. The GelMA hydrogel formulations were characterized using scanning electron microscopy, swelling measurements, biodegradation, and release studies. The biological safety effect of GelMA on human retinal pigment epithelial cells and retinal conditions was verified by in vitro and in vivo experiments. The hydrogel exhibited a low swelling ratio, resistance to enzymatic degradation, and excellent biocompatibility. The swelling properties and in vitro biodegradation characteristics were related to the gel concentration. Rapid gel formation was observed after injection, and the in vitro release study confirmed that TA-hydrogels have slower and more prolonged release kinetics than TA suspensions. In vivo fundus imaging, optical coherence tomography measurements of retinal and choroid thickness, and immunohistochemistry did not reveal any apparent abnormalities of retinal or anterior chamber angle, and ERG indicated that the hydrogel had no impact on retinal function. The GelMA hydrogel implantable intraocular device exhibited an extended duration, in situ polymerization, and support cell viability, making it an attractive, safe, and well-controlled platform for treating the posterior segment diseases of the eye.

## 1. Introduction

Globally, a wide range of diseases of the posterior segment of the eye is responsible for severe vision loss and blindness. These disorders include age-related macular degeneration (AMD), diabetic retinopathy (DR), diabetic macular edema (DME), retinal vein occlusions (RVOs), proliferative vitreoretinopathy (PVR), posterior uveitis, diseases that occur as a result of alterations in the vasculature of the retina, genetic eye disorders, and tumors. Eye structures are endowed with multiple protective mechanisms such as corneal, scleral, and blood-retinal barriers; thus, designing and evaluating drug delivery systems for use in the treatment of posterior segment ocular diseases is extremely challenging. Conventional administration techniques include eye drops and periocular (subconjunctival and retrobulbar) injection. Unfortunately, the cornea and sclera act as anatomic barriers, and the conjunctival, choroidal, and retinal circulation act as functional barriers. They may either block drug penetration into the eye or accelerate its removal. This problem has been largely overcome by intraocular injection when the properties of the administered drug permit [1,2,3,4,5].

However, the intravitreal route of drug administration has limitations related to the need for repeated injections, poor patient compliance, and the risk of serious complications. Frequent injections are known to cause retinal detachment and endophthalmitis. Therefore, methods that allow the localization of drugs within the vitreous cavity followed by sustained release could guarantee several benefits: improving the disease condition, decreasing the frequency of intravitreal (IV) injection, lowering the risk of developing infections, and enhancing patient compliance. In recent decades, multiple drug delivery systems have been developed for IV administration. These include in situ formation of hydrogels, microspheres, poly(lactide) (PLA), and poly(ethylene glycol) (PEG) [6,7]. The selection of appropriate materials for the preparation of long-acting, sustained-release implants is crucial for achieving easy fabrication and functionalization, biocompatibility, sustained drug release, and controllable degradation rates of the implant. Toxicity or inflammatory degradation products restrict the application of drug delivery systems such as chitosan, PLA, and poly(methylidene malonate) (PMM), and these systems are only used for topical or transscleral administration [8,9]. Commercially available implants can be nondegradable (e.g., Iluvien^®^) or degradable (e.g., Ozurdex^®^) [10,11]. PEG is a nondegradable material that has been reported to load triamcinolone acetonide and ovalbumin, a model protein [7]. However, reservoir material remains at the implantation site after drug depletion. In recent years, work on ocular implants has shifted its focus to using biodegradable polymers. The most widely studied commercially available polymer is poly(lactic-co-glycolic acid) (PLGA). PLGA can achieve the prolonged release of several small molecules [6,12,13]. Commercially, PLGA is coextruded with dexamethasone to create Ozurdex^®^, a product that allows several weeks of drug delivery. PLGA is degraded hydrolytically to lactic acid and glycolic acid, which create a highly acidic environment [14] and induce local inflammatory reactions [15].

In situ-forming hydrogels are considered attractive biomaterials that can be engineered to offer several advantages, including less frequent administration, increased patient comfort, and cost reduction. Gelatin methacryloyl (GelMA)-based hydrogels have been used as materials with bioadhesive properties, tissue engineering, and drug delivery [16,17,18]. Photocrosslinking of GelMA obtained from natural hydrogel gelatin presents some interesting advantages [19]. The existence of peptide moieties such as arginine–glycine–aspartic acid allows cell attachment and protease degradation, making GelMA a close mimic of the natural extracellular matrix [20]. On the other hand, GelMA is a versatile material that can be easily modified to possess various biofunctionalities through the encapsulation of molecules such as drugs, growth factors, and cytokines [21]. It has been engineered as an injectable material for the delivery of cells in a minimally invasive manner [22]. Considering the versatile properties of GelMA, we believe that GelMA is a promising material for developing systems for sustained drug release.

In this study, TA based on gelatin methacryloyl (GelMA) was prepared for use as an intraocular delivery system, and its in vitro/ex vivo properties were evaluated regarding their possible application in the prolonged vitreous release of TA. The swelling behavior, in vitro biodegradability, and cytotoxicity of the hydrogels were evaluated. Given its easy injectability, biosafety, and long-lasting nature, this platform can potentially be applied to deliver drugs used to treat a plethora of posterior eye diseases and reduce side effects of repeat injection.

## 2. Result

### 2.1. Properties of GelMA-TA

GelMA hydrogels were formed by the photochemical reaction of methacryloyl units on GelMA chains and activated by LAP, which absorbs visible light at a wavelength of 405 nm (Figure 1A). Light at 405 nm falls within the safer UV range (≈405 nm), preventing retinal damage caused by UV light exposure. The spontaneous gelation speed was 1 min. GelMA hydrogels without TA showed a lighter pale-yellow color after solidification, and the colors became deeper as the GelMA concentration was increased (Figure 1B).

The GelMA-drug delivery system (DDS) samples used in in vitro assays were manufactured using a mixed technique. TA particles were homogenized so that they entered each pore of the material (Figure 1B SEM image). The GelMA hydrogel demonstrates a cross-linking porous architecture. The average pore diameter of different concentrations was measured from scanning electron microscopy images. With increasing GelMA concentration, the pore size of the hydrogel decreased, indicating a denser structure. The apparent pore size decreased from 17.0 ± 3.35 μm to 10.7 ± 2.75 μm as the GelMA concentration was increased from 10% (*w*/*v*) to 20% (*w*/*v*). The pore size diameter was approximately 10–100 μm, which is suitable for the attaching small particles and cells [23,24].

The swelling rate is a crucial indicator for judging the advantages and disadvantages of drug delivery systems. A lower swelling rate is beneficial for materials that are to be used for vitreous injection. In terms of the application of IV drugs, a low swelling rate can help avoid an increase in ocular volume. Our results showed that the concentration of GelMA used in the preparation of the hydrogels directly affected the swelling rate (Figure 2A–D). With increasing GelMA concentration, the swelling rate of the hydrogels decreased, and the weight and volume of the hydrogel reached equilibrium after 8 h and 12 h, respectively. G5 (5%GelMA) had the highest swelling rate (31.69 ± 1.43%), while G20 (20%GelMA) had the lowest (14.57 ± 0.87%). Compact materials absorb less water and have lower swelling rates.

Enzymatic degradation is another major factor that affects the rate of drug release. To investigate the protease-mediated degradation of GelMA and its effect on drug release, we incubated GelMA in a collagenase solution. Changes in the wet weight of GelMA were recorded and used to calculate the rate of degradation of the hydrogel. As shown in Figure 2E,F, G5 degraded rapidly within 12 h (40%), while G20 showed only 20% degradation after 72 h.

### 2.2. Ex Vivo Permeation Study

We evaluated the hydrogels and characterized drug release in PBS to determine diffusive release kinetics in vitro release conditions. The profile of TA release from the GelMA-TA hydrogels in PBS is shown in Figure 3.

To evaluate the effects of hydrogel’s concentration on drug release, two different concentrations of GelMA (10% and 20%, designated G10 and G20, respectively) were used to encapsulate the same dose of TA (1 mg). Our study compared in vitro release profiles of G10 +1 mg TA and 1 mg TA suspension. First, 88.23 ± 1.94% TA was slowly dissolved from the 1 mg TA suspension on the 30 th day of incubation, compared with 42.42 ± 1.66% released from G10 + 1 mg TA (Table 1). On the 90th day of incubation, 97.51 ±1.99% TA was released from G10 + 1 mg TA (Figure 3C). This means that the implant can release TA over a period that is substantially longer than 90 days. The time taken to reach 50% release (t50, release) was 14 d for 1 mg suspension, 37 days for G10 + 1 mg TA, and 46 days for G20 + 1 mg TA. On the other hand, the mean TA release from G10 + 1 mg was 14.14 ± 2.11 μg, a slight increase compared to the 11.71 ± 1.78 μg observed for G20 + 1 mg, but the total cumulative release at 30 d decreased from 42% to 35%. The loose structure of the G10 hydrogel favored a faster release and higher cumulative release of TA. GelMA-DDA caused a noticeably decreased release, demonstrating that release can be tailored by adjustment of the concentration of GelMA used in TA encapsulation (Figure 3).

To assess the effect of the dose parameter on diffusive drug release in PBS, several doses of TA (1 mg, 2 mg, 4 mg, and 8 mg) were added to the precursor solutions with stirring. As the G10 + TA drug concentration was increased from 1 mg/mL to 8 mg/mL, the peak release increased from 16.50 ± 0.98 μg to 75.76 ± 8.33 μg (Table 1). In vitro release level of TA from 1 mg TA suspension was approximately the same as G10 + 4 mg TA hydrogel, the mean release of the drug exhibited sustained release at 28.67 ± 13.31 for 1 mg TA suspension vs 36.73 ± 9.97 for G10 + TA 4 mg. The increase in TA release corresponded to the increased TA load dose, whereas the TA dose was a significant parameter that allowed us to tailor drug delivery (Figure 3B).

### 2.3. In Vitro Assessment of Cytocompatibility

We further performed in vitro cell biocompatibility studies in 2D-cell cultures of HRPE cells. It enabled us to consider the effects of the structural properties of GelMAs with different crosslinking densities on the cellular biocompatibility of the hydrogel. We used a standard live/dead assay to measure cell viability by determining the percentage of live cells remaining after seeding in medium-leached (Figure 4A) and 2D (Figure 4B) cultures. The results demonstrated that cell viability remained >90% after 1, 3, and 5 days in GelMA + TA leached and 2D cultures (Figure 4A,B). In the in vitro study, the effect of GelMA + TA on HRPE cell migration was analyzed. Cell migration was measured using wound healing assays. As Figure 4C shows, gap closure by migrating cells did not obviously differ from that in the control group during incubation of the cells in GelMA + TA leaching medium for 10 h or 24 h after scratching. We tested the stimulatory effect of GelMA + TA in primary cultured RPE cells. As shown in Figure 4D, TA stimulation induced IL-10 upregulation and TGF-β2 downregulation in the cells, indicating that TA affects the anti-inflammatory response in primary RPE cells. Furthermore, 8 min of exposure to 405 nm light induced Nrf2 expression and inhibited H_2_O_2_-mediated damage in primary cultured RPE cells (Figure 4E).

### 2.4. In Vivo Release of TA from Implants

In vivo, a functional assessment of the retina was performed by electroretinography (ERG). Rabbits’ eyes that had been injected with 10% GelMA hydrogel with or without 1 mg TA showed only slightly low amplitudes of the DA 0.01 b-wave and LA 30 Hz flicker at 7 d and recovery by 2 months. At 2 months, there were no significant changes in DA 0.01 b-waves, DA 10 a-waves or b/a ratios, LA 3 b-waves or b/a ratios, or 30 Hz flicker ERG amplitudes in the eyes injected with G10 and G10 + 1 mg TA hydrogel compared to the unoperated eyes (Figure 5).

As assessed by slit lamp examination and fundus evaluation, rabbits whose eyes had been implanted with G10 or G10 + 1 mg TA hydrogels showed no significant inflammation in either the anterior or posterior segment of the eye (columns (i–iii) in Figure 6A–C) and had normal IOP. Furthermore, 10% GelMA demonstrated superior long-term biocompatibility (2 months) in vivo; the injected eyes presented clear corneas, absence of cataracts (Figure 7(Bi)), normal IOP (between 6 and 15 mmHg, Figure 8C), and normal retinal thickness and choroid thickness as determined by both SD-OCT and H&E histology (Figure 7(Biii,Biv); Figure 8A,B). Compared with the non-operated group, the eyes of the G10 and G10 + 1 mg TA groups displayed no disorganized microstructure, apparent inflammatory cells, hemorrhage, or edema two months after injection of the hydrogel (Figure 7A–C). No inflammatory cells were found in the anterior chamber angle in any of the sections examined (Column (v) in Figure 7A–C). These results indicate that the functions of the entire layer of neuroretinal tissue were normal in GelMA-injected eyes. Furthermore, normal IOP values were observed in eyes injected with G10 and G10 + 1 mg TA for 2 months after injection (Figure 8C).

## 3. Discussion

The data presented in this study show that the GelMA hydrogel has several unique functions. First, it can be delivered by a simple injection and cured in 60 s at 37° under 405 nm visible light after injection. It offers a safe, biodegradable implant that can be placed in a posterior vitreous location and acts as a controlled drug release system. This hydrogel, therefore, has the potential to provide a new class of biomaterials suitable for ophthalmic applications.

Clinically, easy injectability is a critical factor in intraocular drug application. In general, conventional pellet-shaped implants require an expensive engineered delivery applicator that is invasive because it usually requires the use of large-gauge needles that must be placed in a region of the sclera that is uncomfortable for the patient [25,26,27]. GelMA that allows in situ sol–gel transition could be used to develop injectable materials that can be delivered locally in a minimally invasive and cost-effective manner. After injection in the soluble state using a standard 23 G or 25 G instrument, it gels within minutes after photocrosslinking with 405 nm light. We tested the expression of the oxidative stress-related gene Nrf2 and its dependent gene HO-1 to characterize oxidative stress following the application of 405 nm light. Our results did not show any increase in oxidative stress in HRPE cells after exposure to visible light at a wavelength of 405 nm for 4 min. Eight minutes of light exposure can induce oxidative stress that inhibits the compensatory upregulation of Nrf2. The photocurability of the material allows temporal and spatial control of the reaction with the assistance of a photoinitiator. From a clinical viewpoint, the most significant advantage of the light-curing adhesive system is that it provides surgeons with sufficient working time to properly position the graft before using light to gelatinize it.

Various drug delivery systems (DDSs) have been explored in the scientific literature, including membranes [28], nanoparticles [29], liposomes [30], and hydrogels [31]. Hydrogels are a particularly interesting class of materials for use as DDSs and have been extensively used in many branches of medicine and tissue engineering. We focused on the impact of drug loading on the equilibrium swelling and the GelMA-DDS mesh size. Mesh size is the primary physical parameter that controls drug diffusion [31]. SEM images of the GelMA hydrogels at different GelMA concentrations are shown in Figure 1. As shown in the images, the crosslinked network forms a porous three-dimensional structure, and the pores are uniformly distributed in the structures. The pore size decreased as the GelMA concentration increased, and the increased density of the hydrogel network hindered the release of the loaded drugs from the hydrogel. The porous structure provides appropriate channels for medium flow and drug transport, making the hydrogel favorable for drug release. Other studies have demonstrated that increasing the GelMA concentration decreases the hydrogel mesh size [32,33], leading to higher stiffness and lower permeability [34].

Tests of a material’s mechanical properties are crucial when designing implantable hydrogel-DDSs that possess stability and degradability. Figure 2E shows that the enzymatic resistance of hydrogels improved with increasing GelMA concentration. The degradation of GelMA was slow, requiring approximately 100 h for G5 and 150 h for G20. During the degradation process, the hydrogel gradually shrinks and eventually disappears instead of breaking into several pieces. It is important for intraocular applications, as small fragments may obstruct the trabecular meshwork.

Figure 1 shows that with increasing GelMA concentration, the hydrogels show a denser structure and have a low swelling rate. In terms of their use to deliver IV drugs, a low swelling rate can help avoid an increase in ocular volume. However, increasing the compactness of the GelMA leads to higher stiffness and lower injectability. Therefore, we chose a suitable concentration, G10, to achieve easy injectability and a low swelling rate. The hydrogel tensile strength and strain at break averaged 18–45 kPa and 34–48%, respectively. Young’s modulus of the hydrogels prepared at different concentrations of GelMA ranged from 20 KPa (G5) to 98 KPa (G20) in our previous study [17]. Overall, the expected range of mechanical properties observed for 10% GelMA is consistent with that reported in other studies [35,36].

Triamcinolone acetonide (TA) is a synthetic corticosteroid structured as 9-fluoro-11b,16a,17,21-tetrahydroxypregna-1,4-diene-3,20-dione cyclic 16,17-acetal with acetone. TA is used extensively in treating ocular diseases characterized by inflammation, edema, and neovascularization. To understand the effect of GelMA in vitro drug release, TA was used as a model drug and loaded into hydrogels prepared at different GelMA concentrations. We evaluated the release of the drug in PBS to obtain diffusive release kinetics in vitro release conditions. The release kinetics of TA from GelMA were assessed over 90 d. Figure 3 shows that the drug release rate was dependent on the hydrogel concentration and that TA release rate decreased with increasing gel concentration.

Intravitreal injection of 1 mg/0.1 mL TA suspensions is the most popular choice for ophthalmologists. Previous studies showed that 1 mg TA release in vivo was 99.1 ± 0.4% complete after just 21 days, and with a half-life of 15.4 ± 1.9 days [37,38]. Increasing the dose of intravitreal TA could prolong the effects of the drug on the retina, the calculated half-lives of intravitreal TA were 24 days for the 4 mg dose and 34 days for the 8 mg dose [39]. Unfortunately, the high dosage was associated with an increased risk of steroid-related IOP rise [40,41]. In vitro release level of TA from 1 mg TA suspension was approximately the same as G10 + 4 mg TA hydrogel. The mean release of the drug exhibited sustained release at 28.67 ± 13.31 μg for 1 mg TA suspension vs 36.73 ± 9.97 μg for G10 + TA 4 mg. The release concentrations of 1 mg TA doses decreased gradually with time, and the peak concentration was the initial concentration. The initial concentration was 6-fold the amount released at 30 days. However, in GelMA platform, there is no apparent initial release burst, preventing an undesired high initial dosage. More important, T50, release for G10 + TA 4 mg was 60 days compared to 14 days for 1 mg TA suspension. We observed that GelMA reservoirs achieved more drug release as the drug dose was increased, demonstrating the dose-dependent characteristics of drug release. The use of a controlled release rate of TA could reduce the risk of cytotoxicity.

An appropriate degradation rate of the material is crucial in selecting materials for long-acting, sustained-release implants. The first generation of ocular drug delivery systems commonly use nonbiodegradable implants. Retisert^®^ and Vitrasert^®^, which are drug tablets coated with nonbiodegradable polymers, are two examples. However, after depletion of the drug, the drug reservoir requires surgical removal or is allowed to remain at the implantation site, increasing the likelihood of complications. In recent years, interest in using biodegradable polymers as ocular implants has grown. Among such biodegradable polymers, PLGA is the most widely discussed in the literature and is the most widely commercially available [12,13]. In the commercial product Ozurdex^®^ (Allergan, Dublin, Ireland), PLGA is coextruded with dexamethasone to achieve 3 months of drug delivery. A limitation of PLGA is that it is degraded to lactic acid and glycolic acid, which create a highly acidic environment [14] and cause local inflammatory reactions [15]. Previous studies have demonstrated that GelMA hydrogels consist of gelatin backbones and that their degradation products consist of Amide A, Amide I, Amide II, and Amide III [42]. Thus, to imitate the in vivo environment, we also evaluated the degradation behavior of GelMA hydrogels in collagenase solution by monitoring the percentage of residual hydrogel mass as a function of time (Figure 2E,F). As expected, more rapid loss of mass was observed for the GelMA hydrogels with lower concentrations. As presented in Figure 2E,F, the enzymatic resistance of hydrogels was improved as the GelMA concentration was increased, similar to the swelling rate trend. The degradation time of G5 was the shortest (approximately 100 h), while that of G20 was the longest (approximately 150 h). The degradation time of 50 μL G10 in rabbit eyes was 3 months. GelMA hydrogels are based on gelatin, an inexpensive denatured form of collagen susceptible to enzymatic degradation. Clinically, this is a major advantage of its use in ocular implants.

The potential cytotoxicity of the Id hyIrIgels is important for Iheir use in intravitreal ocular drug delivery. GelMA has been widely used in biomedical applications such as drug carriers and cell ECM and showed low cellular cytotoxicity [21,43]. This study measured the viability of RPE cells by live/dead assays to evaluate the cytotoxicity of the GelMA and TA mixture hydrogels prepared in this study. The cells were incubated for 1, 3, or 5 d with hydrogel leachates, during which time the medium was not changed. The results show that GelMA hydrogels are promising ocular drug carriers that may not affect the viability of cells.

Testing the ocular tolerability of a drug delivery system designed for ocular instillation is extremely important. The complications are associated with intravitreal triamcinolone therapy include secondary ocular hypertension in approximately 40% of the eyes injected, medically uncontrollable high intraocular pressure leading to secondary open-angle glaucoma and requiring antiglaucomatous surgery in approximately 1% to 2% of the eyes injected, posterior subcapsular cataract and nuclear cataract leading to cataract surgery in approximately 15% to 20% of elderly patients within 1 year after injection, postoperative infectious endophthalmitis at a rate of approximately 1:1000, noninfectious endophthalmitis, perhaps due to a reaction to the solvent agent, and pseudoendophthalmitis in which triamcinolone acetonide crystals appeared in the anterior chamber of the eye [44].

As previously mentioned, the most important adverse reaction to administration of intraocular TA (IVTA) is elevated IOP. In several studies, white particles were found in the anterior chamber and angle after IVTA injection, and these were correlated with an IOP rise [45,46]. It might be related to the spillover of TA into the anterior chamber and a reduction in the outflow capacity of the trabecular meshwork. To examine the in vivo drug side effects, the effects on IOP of administration of 1 mg TA mixed with 10% GelMA (*w*/*v*) were evaluated and compared with the effects of administration of the conventional solution (suspension). There were no significant differences in IOP among the three groups during the 60-day study period. In this work, intraocular tissues were evaluated via OCT and H&E staining to determine the levels of inflammation after implantation of the hydrogel. It could pave the way for further clinical research. Compared with normal control eyes, full-field ERG shows both scotopic and photopic function at three months in rabbits. No abnormal changes were found on ERG examination 2 months after implantation of GelMA in the rabbits’ eyes (Figure 5). The GelMA and GelMA + TA groups showed no disorganized microstructure, apparent inflammatory cells, hemorrhage, or edema at one month after implantation, and no inflammatory cells were found in the ciliary bodies in any of the sections examined (Figure 7A–C). Collectively, the function and structure examinations further proved that the novel hydrogel could be an excellent choice for use in vitreous implant drug applications.

A limitation of our study is that we did not measure TA concIntrations in rabbits to demonstrate the pharmacokinetic features of the hydrogels in vivo. However, TA is a well-studied drug, and drugs in the vitreous can be eliminated through the retina-choroid layer that surrounds the vitreous and/or by the aqueous humor outflow pathways in the anterior of the eye. Furthermore, GelMA has a binding affinity for growth factors, suggesting a potential role for this material as a biological medicine carrier [47].

## 4. Materials and Methods

### 4.1. Materials

GelMA (EFL-GM-90, 90% graft degree) and lithium phenyl-2,4,6- trimethybenzoylphosphinate (LAP) were purchased from Suzhou Intelligent Manufacturing Research Institute (Suzhou, Jiangsu, China). All cell culture-related reagents were purchased from Gibco BRL (Grand Island, New York, NY, USA) and dispase (Roche, Indianapolis, IN, USA). Fluorescent F-actin/DAPI staining fluorescent and LIVE/DEAD assay kits were purchased from Invitrogen (Thermo Fisher Scientific, Shanghai, China). PrimerScript RT Master Mix and SYBR Green Supermix were obtained from TaKaRa Biotechnology (Kusatsu, Japan). Triamcinolone acetonide (TA) was purchased from Zhejiang Xianju Pharmaceutical Co., Ltd. (Taizhou, Zhejiang, China). The anti-RPE65 antibody was purchased from Abcam (Shanghai, China). Anti-rabbit IgG (4412, Alexa Fluor 488 conjugate) antibodies were purchased from Cell Signaling Technology (CST, Danvers, MA, USA). Collagenase II was purchased from Sigma-Aldrich (St. Louis, MO, USA).

### 4.2. Preparation of the GelMA Hydrogels Solutions and TA-Loaded GelMA Solution

LAP was dissolved in PBS to a concentration of 0.25% (*w*/*v*) and heated at 55 °C to ensure complete dissolution. GelMA powder and LAP solution were mixed thoroughly at concentrations of 5%, 10%, and 20% (*w*/*v*). TA was loaded at concentrations of 20 mg/mL, 40 mg/mL, 80 mg/mL, and 160 mg/mL into the prepared GelMA solution by a simple mixing technique. The precursor solutions were incubated at 37 °C for 10 min to form Schiff bases and then photocrosslinked with visible light (405 nm, 30 mW/cm [2]) for 1 min using a light source (Suzhou Intelligent Manufacturing Research Institute, Suzhou, China).

### 4.3. Scanning Electron Microscope Imaging

The morphology and external surfaces of the lyophilized samples were examined by scanning electron microscopy (SEM, EVO18; Zeiss, Jena, Germany). Freezing of the samples was performed at −80 °C for 24 and was followed by lyophilization at −50 °C for 72 h by applying a vacuum of 10^−1^ mbar.

### 4.4. Swelling Measurement

The swelling kinetics of GelMA hydrogels (~500 μL) at various concentrations (5%, 10%, and 20%) were determined at 37 °C. After gel formation at 37 °C for 24 h, the gels were further incubated in PBS (pH = 7.4) at 37 °C. At predetermined time intervals (1, 2, 4, 6, 8, 10, 12, and 24 h), the excess PBS was removed, and the surface water on the hydrogels was blotted gently with filter paper. The gels were then weighed. The swelling ratio was calculated using the following equation, in which swelling ratio = (Wt − Wi)/Wi × 100%, where Wt is the wet weight at a specific time point and Wi is the initial weight before swelling. Every reported value was the average of at least three measurements.

### 4.5. In Vitro Degradation Study of GelMA

The enzymatic degradation of GelMA hydrogels was evaluated using collagenase, as reported elsewhere [48]. Briefly, the GelMA solution (200 μL) was transferred to a PDMS mold (d = 6 mm and th = 3 mm) and polymerized to yield disc-shaped constructs. The discs were then equilibrated overnight in PBS solution. They were then soaked in 5 mL of 0.1 M Tris–HCl buffer (pH 7.4) containing 5 mM CaCl_2_ for 1 h to reabsorb the water. Subsequently, collagenase (Sigma-Aldrich, USA) solution was added to give a final concentration of 1 mg/mL. The collagenase solution was changed every 8 h, and the residue was carefully removed from the solution, gently blotted on filter paper, and weighed. The degradation rate (n = 3) was calculated using the following equation: Lost Weight (%) = 100 (Wt − Wi/Wi × 100, where Wt is the residual wet weight at a specific time point and Wi is the initial wet weight.

### 4.6. In Vitro Drug Release

The in vitro release of TA from hydrogels in PBS (pH 7.4) at 37 °C was measured. First, 1–8 mg amount total TA was loaded into the precursor GelMA solution (200 μL). After the formation of hydrogel, 25 mL phosphate buffer saline (PBS, pH 7.4, 37 °C) containing 10% methanol, which is used to improve the solubility of TA in PBS, was added into the vial as the release medium. The whole experiment setups were placed in a shaking water bath adjusted at a temperature of 37 °C and shaken at a rate of 50 strokes per min. At predetermined time intervals, 2 mL of the sample was withdrawn, and the same volume of fresh PBS was added to the dissolution vessel. The released TA was detected by high-performance liquid chromatography (HPLC, LC-20 A, Shimadzu, Kyoto, Japan) on a C18 reversed-phase analytical column with spectrometric detection at 239 nm. The mobile phase consisted of 525 parts methanol and 475 parts deionized water. The flow rate was 1.0 mL/min, and the separation was performed at 40 °C. The amount of TA present was determined using a standard curve. The amount of drug release was expressed as the cumulative percentage of TA released in PBS. The experiments were performed at least in triplicate for each sample. The average values of the obtained results were calculated and used to plot the in vitro time-release profile.

### 4.7. In Vitro Cell Studies

#### 4.7.1. Cell Cultures

Cytotoxicity studies were conducted using human retinal pigment epithelial cells (HRPE) because this cell line is widely used for cytotoxicity testing in the literature.

A pair of human eyes from a 7-year-old donor was obtained from the eye bank of Zhongshan Ophthalmic Center. The research protocol was approved by the Medical Ethics Committee of Zhongshan Ophthalmic Center, Sun Yat-sen University (Approval No. 2018KYPJ034, Approval Date 1 March 2018), and the tenets of the Declaration of Helsinki were followed throughout the study. RPE cells were harvested under sterile conditions as described below. Briefly, the anterior segment of each eye was removed by cutting around the iris. The opened eye was transferred to a new plate, the neural retina was removed, and the remaining part of the eye was rinsed several times with PBS to eliminate remaining neural tissues, blood, and other residual tissue. The retinal pigment epithelium-choroid was then carefully separated from the sclera and placed face up in a small sterile Petri dish containing 2 U/mL dispase in Ca^2+^- and Mg^2+^-free Hank’s balanced salt solution. After incubation at 37 °C for 1 h with occasional shaking, the supernatant was carefully aspirated and transferred to a sterile centrifuge tube. The sample was then centrifuged at 1000 rpm for 5 min, and the resulting pellet was resuspended gently in Dulbecco’s modified Eagle’s medium (DMEM)/F12 supplemented with 10% fetal bovine serum and 100 U/mL penicillin and streptomycin. The suspended cells were transferred to 25 cm^2^ flasks and cultured in DMEM/F12 containing 10% fetal bovine serum in a humidified incubator at 37 °C and 5% CO_2_. Subculture using trypsinization with trypsin/EDTA solution was performed when the cells reached 80% confluence. Experiments were conducted on cells at passages 3 to 7. For hydrogel leaching medium culture, the hydrogels were immersed in culture medium at a ratio of 1 mL medium per 0.1 g gel and incubated with the cells at 37 °C in a humidified atmosphere of 5% CO_2_ for 24 h. For two-dimensional (2D) culture, the precursor solutions were prepared as described above and filtered through a 0.22-μm filter (Merck Millipore, Burlington, MA, USA). The sterile solutions were then added to tissue culture plates and photocrosslinked in the wells. After three washes with the culture medium, HRPE cells were seeded on top of the hydrogels. The culture medium was replaced every 2 days.

#### 4.7.2. Assessment of HRPEs Cell Viability

The viability of cells was evaluated using a LIVE/DEAD^®^ Viability/Cytotoxicity kit for mammalian cells (Invitrogen™) according to the manufacturer’s instructions. Briefly, cells were incubated with 0.5 μL/mL calcein AM and 2 μL/mL ethidium homodimer-1 (EthD-1) in DPBS for 15 min at 37 °C in the cell culture incubator to allow simultaneous staining of living and dead cells. At 1, 3, and 5 days post-seeding, live (green-stained) and dead (red-stained) cells were imaged using an inverted fluorescence microscope (Observer 7, Zeiss, Germany). The numbers of live and dead cells were quantified using ImageJ software. Viability was then calculated using the following equation: Cell viability (%) = Living cells/ (Living cells + Dead cells) × 100%.

#### 4.7.3. HRPE Migration Function

Cell migration was measured using wound healing assays. For the wound healing assay, HRPE cells (5 × 10^5^ cells/mL) were seeded in 6-well plastic culture plates. When the cells reached approximately 80–90% confluence, straight scratches were made in the cell monolayer using a micropipette tip. The plain culture medium and containing the G10 + TA1 mg leaching culture medium were then added to the wounded cells. At intervals thereafter, the width of the scratch width was quantified using ImageJ software, and the percentage of the gap covered by migrated HRPE cells was taken to indicate the migration rate. The data used in the quantification were obtained in three independent experiments.

#### 4.7.4. UV Radiation Procedure

HRPE cells were exposed to blue visible light (405 nm, 30 mW/cm^2^) for 30 s, 1 min, 2 min, 4 min, and 8 min at a distance of 1 cm, followed by culturing in basal medium.

#### 4.7.5. Quantitative Real-Time PCR (qRT-PCR)

Treated cells were collected, total RNA was extracted using TRIzol reagent (Invitrogen, Waltham, MA, USA), and 1 μL of the extracted total RNA was reverse-transcribed for cDNA synthesis using a SYBR PrimeScript™ RT-PCR Kit (Takara, Dalian, China) according to the manufacturer’s instructions. The expression levels of interleukin-6 (IL-6), interleukin-10 (IL-10), nuclear factor E2-related Factor 2 (NrF2), heme oxygenase-1 (HO1), glyceraldehyde-3-phosphate dehydrogenase (GAPDH), transforming growth factor beta 1 (TGF-β1) and transforming growth factor beta 2 (TGF-β2) were measured by qRT-PCR. using a LightCycler480 II Sequence Detection System (Roche, Switzerland). Relative target gene expression was calculated using the ΔΔCt method. The primer sequences were as follows: IL-6, forward: 5′-AAGCCAGAGCTGTGCAGATGAGTA-3′, reverse: 3′-TGTCCTGCAGCCACTGGTTC-5′; IL-10, forward: 5′-TAATTTATCTTGTCTCTGGGCTTGG-3′, reverse: 3′-AAGTGGTTGGGGAATGAGGTT-5′; TGF-β1, forward: 5′-CGCATCCTAGACCCTTTCTCCTC-3′, reverse: 3′-GGTGTCTCAGTATCCCACGGAAAT-5′; TGF-β2, forward: 5′-TTACACTGTCCCTGCTGCACTT-3′, reverse: 3′-GGTATATGTGGAGGTGCCATCAA-5′; NrF2, forward: 5′-CGGTATGCAACAGGACATTG-3′, reverse: 3′-ACTGGTTGGGGTCTTCTGTG-5′; HO1, forward: 5′-AAGATTGCCCAGAAAGCCCTGGAC-3′, reverse: 3′-AACTGTCGCCACCAGAAAGCTGAG-5′; and GAPDH, forward: 5′-GCACCGTCAAGGCTGAGAAC-3′, reverse: 3′-TGGTGAAGACGCCAGTGGA-5’.

### 4.8. Animal Studies

All experimental protocols, including experimental procedures and the transport and care of the animals, complied with the Association for Research in Vision and Ophthalmology Statement for the Use of Animals in Ophthalmic and Vision Research and the guidelines provided by the Animal Care and Use Committee of Zhongshan Ophthalmic Center (Guangzhou, Guangdong, China).

#### 4.8.1. Intravitreal Gel Injection

Eight- to twelve-week-old male New Zealand White rabbits (Guangzhou Huadu Hua Dong Xin Hua Experimental Animal Farm, Guangdong, China) were used in this study. The rabbits were randomly divided into three groups (a GelMA group, a GelMA + TA group, and a control group), with 3 animals in each group. The rabbits were housed under standard conditions (25 °C, relative humidity 50%) in the animal facilities of the Zhongshan Ophthalmic Center (Sun Yat-Sen University) and given free access to food and water. For hydrogel injection, the rabbits were placed under general anesthesia by intravenous injection of 2% pentobarbital sodium (30 mg/kg). After intramuscular injection of xylazine solution (0.5 mL/kg body weight), topical anesthesia (0.5% Alcaine (Alcon)) was administered to reduce the animals’ discomfort. While anesthetized, the rabbits were kept on a heating pad to enable them to maintain their body temperature. The pupils were dilated with topical 1% tropicamide (Alcon). The ocular surface and surrounding tissue were sterilized with diluted iodophor solution followed by saline. Using C-ring and a coverslip, the posterior eye chamber and the retina could be visualized under a surgical microscope, allowing real-time monitoring of the procedure. An incision was then made into the vitreous using a 25-gauge trocar (Alcon) without disturbing the posterior capsule, lens, or retina. Fifty microliters of prepared precursor solutions were injected into the vitreous through a sterilized 1 mL syringe. The injected precursor solutions were then photopolymerized by exposure to 405 nm visible light for 1 min. Finally, an Elizabeth collar was used to prevent the animal from scratching its eyes. The postoperative regimen included tobramycin and dexamethasone eye ointment (Tobradex, Alcon, Fort Worth, TX, USA) once daily for one week.

#### 4.8.2. Slit-Lamp, Color Fundus Photos, and Optical Coherence Tomography Evaluation for Ocular Media and Retinal Tissue

Slit lamp biomicroscopy (Topcon system) was performed to evaluate anterior chamber (AC) transparency and possible pathological changes. Color fundus photography was performed using a Topcon Fundus Camera (Topcon system) at designated time points to evaluate the retina and detect any changes in the hydrogel. Images showing the structure and thickness of the retina and the choroid tissue were obtained by spectral-domain OCT (Heidelberg Engineering) and used in the evaluation.

#### 4.8.3. Assessment of Retinal Function by Electroretinography (ERG)

ERG was recorded before (baseline) and 1 week, 2 weeks, and 4 weeks after injection of GelMA hydrogel into the vitreous to determine whether GelMA is harmful to the electrophysiological function of the retina. In each rabbit, the maximum dark-adapted b-wave amplitudes were recorded. The ratio of the amplitudes measured before and after injection of the eye was used in the analysis to avoid possible artifacts caused by individual variation in ERG amplitude.

#### 4.8.4. Intraocular Pressure (IOP) Measurements

The IOP of rabbits was measured using an Applanation Tonometer (Tono-Pen Avia; Reichert, Inc., Depew, NY, USA) by gently and vertically touching the center of the cornea after administration of a topical anesthetic. Each measurement was performed at 10–11 a.m. by the same technician. Baseline IOP was obtained after anesthesia but before the operation. IOP measurements were obtained again immediately after surgery and at each sampling time point thereafter. Ten data points per eye were measured and averaged.

#### 4.8.5. Histopathological Examination

One rabbit in each group was euthanized 2 months after injection. The rabbit’s eyeballs were removed and immediately fixed in 4% paraformaldehyde. The eyes were then embedded in paraffin, cut, and dewaxed. Sections (10 μm in thickness) were subjected to hematoxylin and eosin (HE) staining. HE staining of the retinal sections was imaged using Tissue Fax software and analyzed using ImageJ. In each retinal section, the number of endothelial cells that broke through the inner limiting membrane was counted.

### 4.9. Statistical Analysis

All data are presented as mean ± standard deviation. The data were analyzed by one-way ANOVA or Student’s *t*-test using GraphPad Prism 9 (GraphPad Software, San Diego, CA, USA). A value of *p* < 0.05 was considered statistically significant.

## 5. Conclusions

In this work, TA was released from GelMA + TA in a sustained manner after injection into the vitreous. The safe delivery and in situ polymerization of GelMA hydrogels crosslinked by visible light make them remarkably suitable for intravitreous injection. The hydrogel showed excellent biocompatibility both in vitro and in vivo for 2 months after injection into rabbits’ eyes, and it exhibited sustained release of the drug without ocular complications. Therefore, GelMA hydrogel has great potential for use in the posterior vitreous administration of drug-loaded systems.

## Figures and Tables

**Figure 1 ijms-24-04957-f001:**
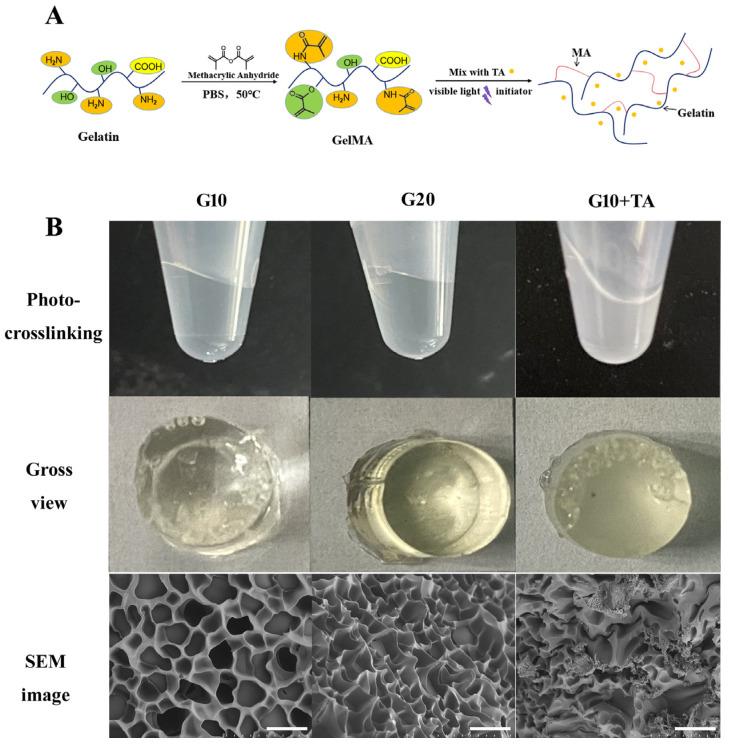
Fabrication and observation of gelatin methacrylate (GelMA) hydrogels at 10% and 20% GelMA, and 10% GelMA with 1 mg/50 μL TA. (**A**) Schematic route of the synthesis of GelMA + TA hydrogel. (**B**) Gross comparison of hydrogel formation and SEM images of hydrogels prepared at various GelMA: TA ratios. Scale bar: 25 μm.

**Figure 2 ijms-24-04957-f002:**
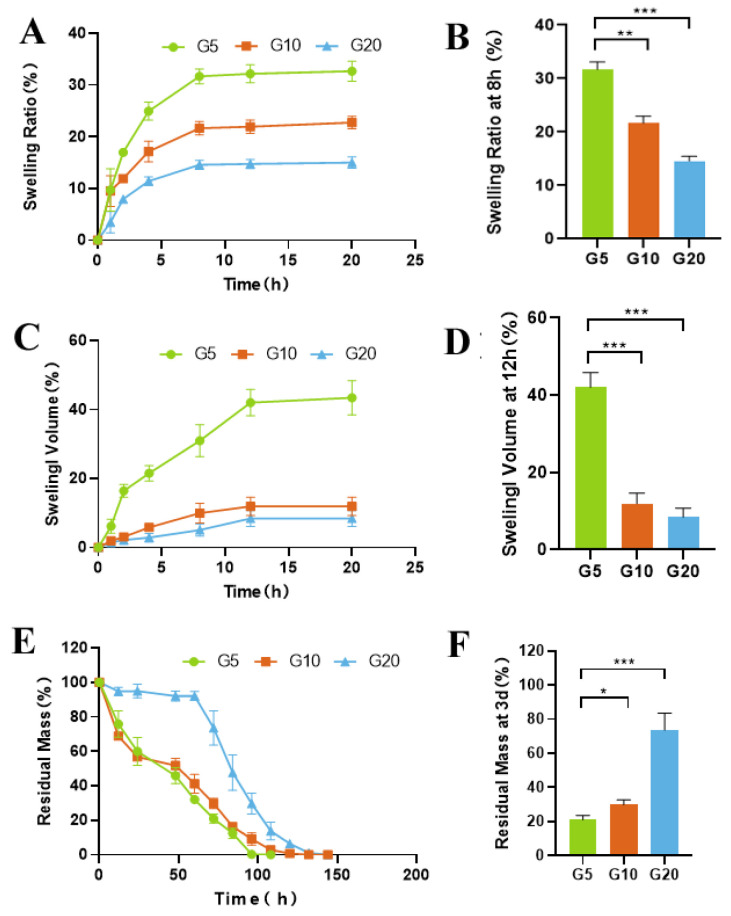
In vitro characterization of the GelMA hydrogels. (**A**) Swelling ratios of the hydrogels at different time points in PBS at 37 °C. (**B**) Swelling ratios of the hydrogels after incubation in PBS at 37 °C for 8 h. (**C**) Volumes of the hydrogels after incubation in PBS at 37 °C for various amounts of time. (**D**) Swelling volume of the hydrogels after incubation in PBS at 37 °C for 12 h. (**E**) Degradation curve of the hydrogels in the presence of collagenase (1 μg/mL) in PBS at 37 °C. (**F**) Degradation of the hydrogels after incubation in PBS containing 1 μg/mL collagenase for 3 d. (* *p* < 0.05, ** *p* < 0.01, *** *p* < 0.001; *n* = 3).

**Figure 3 ijms-24-04957-f003:**
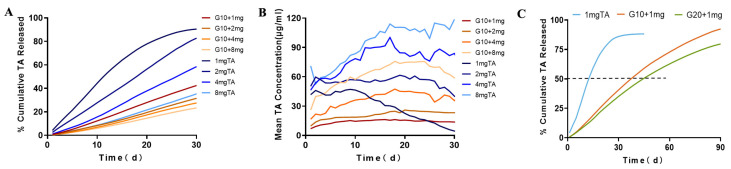
In vitro release profiles for the release of TA from GelMA + TA and TA suspension. (**A**) Cumulative TA release against varying TA-loading concentration from 10% GelMA and suspension over a period of 30 d. (**B**) Mean concentration of TA release against varying TA-loading concentration from 10% GelMA and suspension over a period of 30 d. (**C**) Cumulative TA release against 1 mg TA-loading concentration from 10%, 20% GelMA and suspension during incubation at 37 °C for 90 d with constant shaking at 50 rpm (*n* = 3).

**Figure 4 ijms-24-04957-f004:**
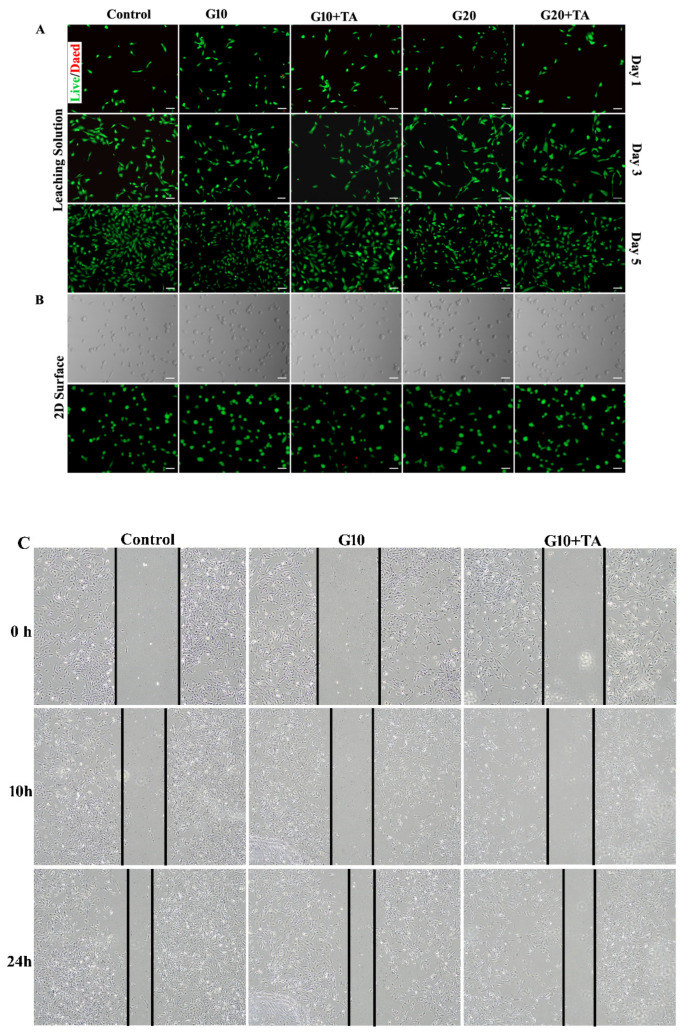
Assessment of the in vitro cytocompatibility of the GelMA + TA hydrogels. Representative bright field and live-dead images of HRPE cells that were seeded on tissue culture plates in complete medium and GelMA + TA leaching medium (10%, 20% *w*/*v* load with or without 1 mg TA) obtained on Days 1, 3, and 5 (**A**) and on the surface of the GelMA + TA hydrogel after 1 day of seeding (**B**) are shown. (**C**) The effect of GelMA + TA leaching medium on cell migration was determined using wound-healing assays. (**D**) qRT-PCR analysis of the relative mRNA expression levels of IL6, TGFβ-1, TGF-β2, and IL-10 mRNA in HRPE after culture in complete medium and in 10% GelMA ± TA leaching medium for 2 days. (**E**) HRPE cells were exposed to 405 nm light for the indicated times (2, 4, and 8 min), and IL6, TGFβ-1, TGF-β2, IL-10, Nrf2, and HO-1 mRNA expression was assayed by qRT-PCR. Scale bar: 100 μm. (** *p* < 0.01, *** *p* < 0.001).

**Figure 5 ijms-24-04957-f005:**
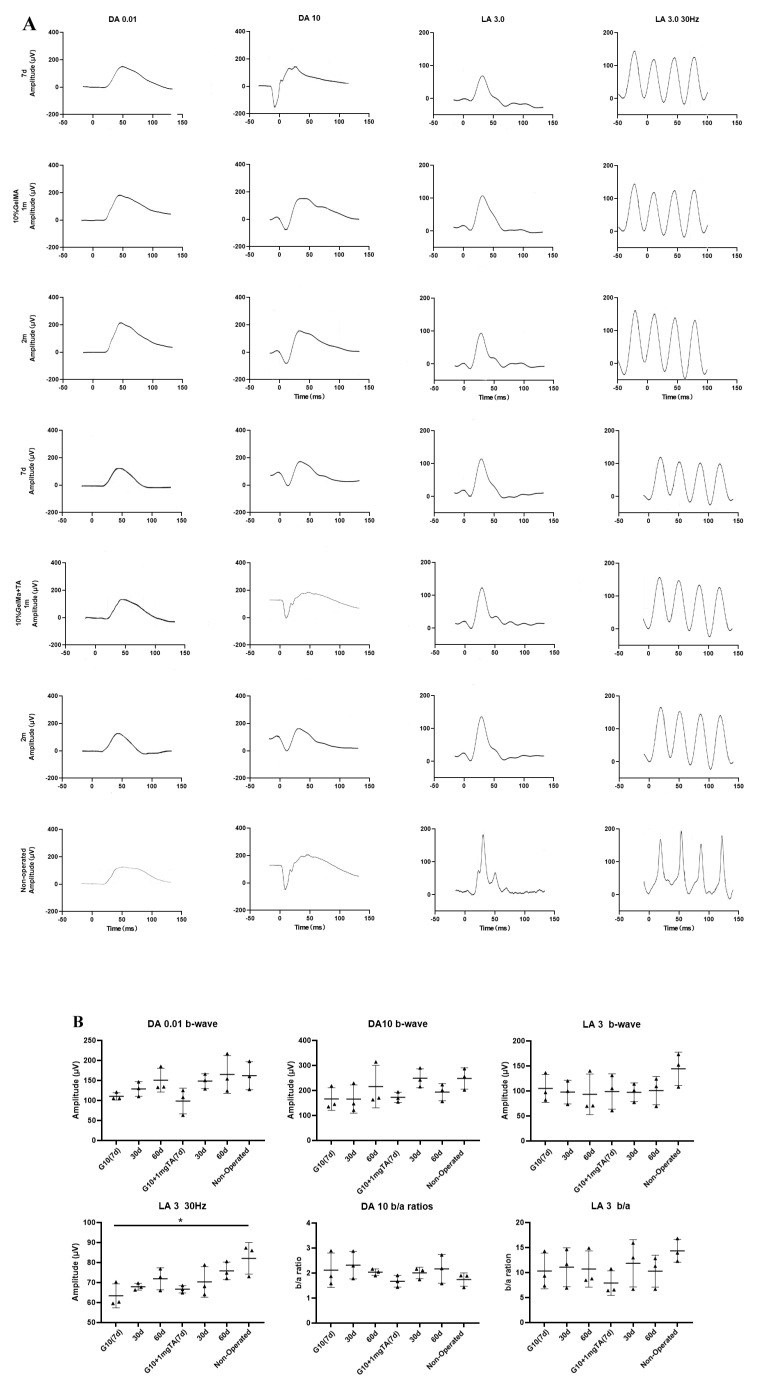
Functional assessment of rabbit retina by ERG. (**A**) In rabbit eyes injected with GelMA hydrogel, there was no significant dysfunction or reduction in scotopic (DA) or photopic (LA) measured by ERG at 7 d, 1 month, or 2 months. In contrast, in rabbit eyes injected with G10 + 1 mg TA for 1 month, there was only a mild reduction in b/a ratios in the inner retinal layer compared to normal eyes. (**B**) Scatter plots showing DA 0.01, DA 10, DA 10 b/a ratios, LA 5, LA 3 b/a ratio, and LA 3 30 Hz in the unoperated eyes and in the G10, and G10 + 1 mg TA hydrogel-injected eyes of six rabbits. Statistical analysis was performed in C using ANOVA with Dunnett’s post hoc test. The data are presented as mean ± standard deviation (* *p* < 0.05).

**Figure 6 ijms-24-04957-f006:**
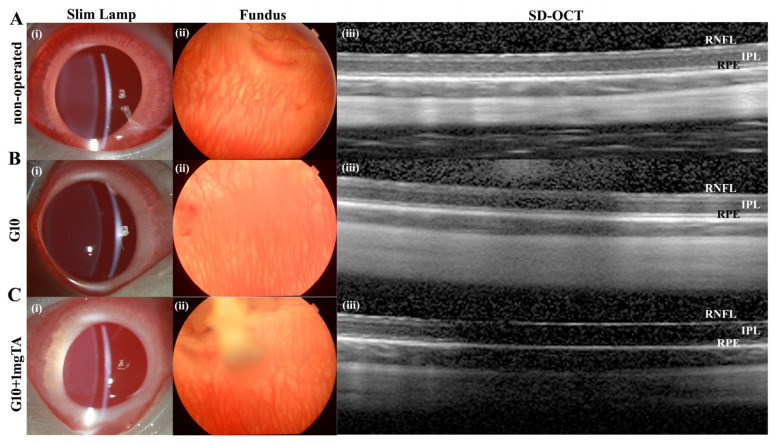
In vivo imaging and ex vivo retinal analysis of rabbits 7 d after implantation of GelMA hydrogels. (**A**–**C**), Column (**i**): slit-lamp images of operated BSS controls and G10- and G10 + 1 mg TA-injected eyes, showing no significant inflammation or cataract formation. Column (**ii**): images of the fundus showing normal appearance of the optic disc and normal vessel morphology in all groups. Column (**iii**): SD-OCT images obtained in all four groups revealed no reduction in retinal thickness from the RPE to the RNFL.

**Figure 7 ijms-24-04957-f007:**
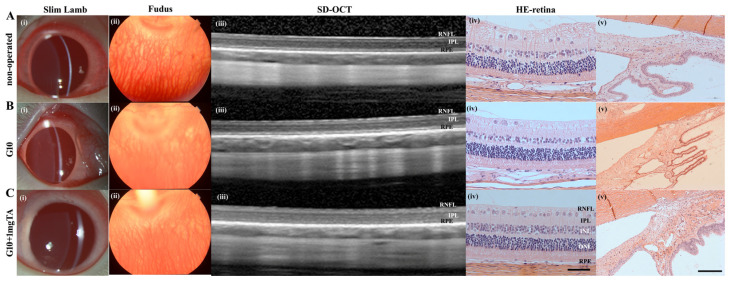
In vivo imaging and ex vivo retinal analysis of rabbits 2 months after implantation of GelMA hydrogels. (**A**–**C**), Column (**i**): slit-lamp images of operated BSS controls and G10 and G10 + 1 mg TA-injected eyes, showing no significant inflammation or cataract formation. Column (**ii**): images of the fundus revealing normal optic disc appearance and vessel morphology in all groups. Column (**iii**): SD-OCT images obtained in all four groups revealed no reduction in retinal thickness from the RPE to the RNFL. Column (**iv**): H&E histology analysis was performed in all three groups, and the results were consistent with the OCT findings (scale bars, 50 μm). Column (**v**): H&E histological analysis of the anterior chamber angle (scale bars, 50 μm).

**Figure 8 ijms-24-04957-f008:**
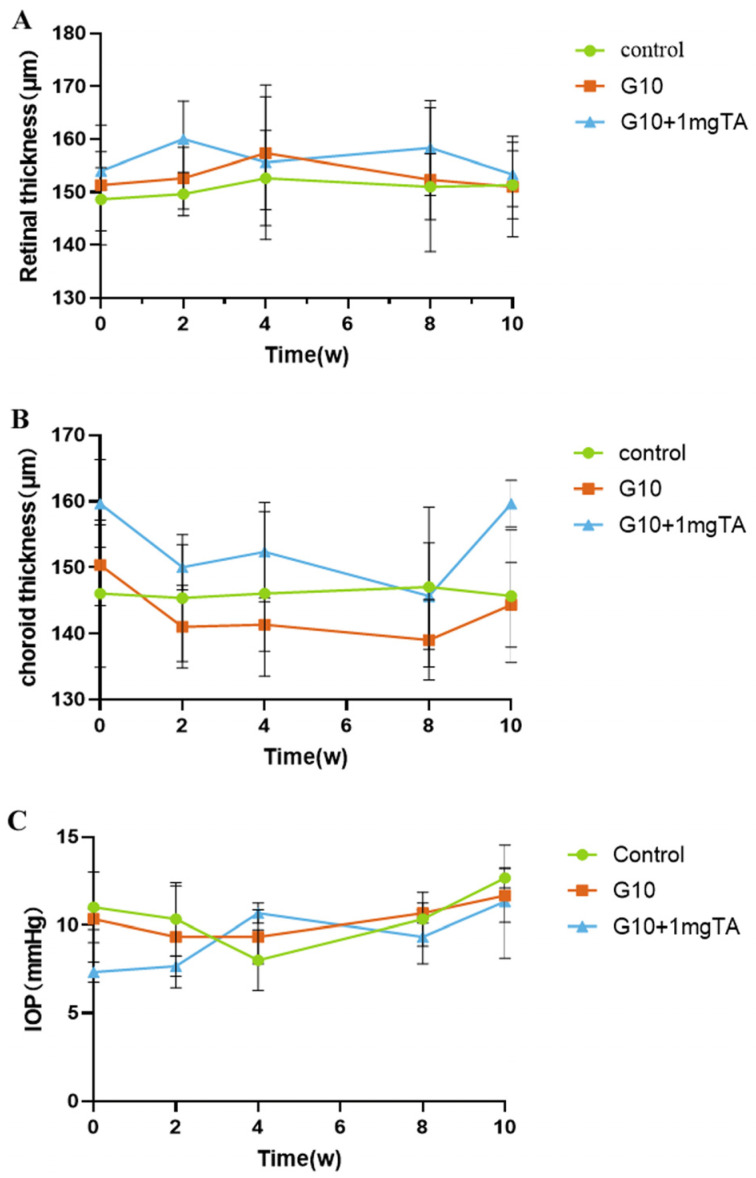
Quantitative analysis of changes in retinal/choroid thickness and intraocular pressure occurring throughout the study. Retinal thickness (**A**) and choroid thickness (**B**) was measured by OCT over the course of the study. (**B**) Choroid thickness was measured by OCT over the course of the study. (**C**) Change in IOP when the GelMA-TA system was injected into rabbits.

**Table 1 ijms-24-04957-t001:** Measured the in vitro triamcinolone acetonide release from GelMA.

Group Denomination	GelMA Concentration (%)	Dose (mg)	Mean Release (μg) of 30 d	Cumulative Release of 30 d	Peak Release (μg)
G10 + TA 1 mg	10	1	14.14 ± 2.11	42.42 ± 1.66	16.50 ± 0.98
G10 + TA 2 mg	10	2	21.01 ± 3.92	31.51 ± 1.18	24.79 ± 2.48
G10 + TA 4 mg	10	4	36.73 ± 9.97	27.55 ± 1.59	45.16 ± 3.12
G10 + TA 8 mg	10	8	62.14 ± 12.69	23.30 ± 1.86	75.76 ± 8.33
G20 + TA 1 mg	20	1	11.71 ± 1.78	35.14 ± 1.29	14.95 ± 1.37
G20 + TA 2 mg	20	2	18.12 ± 4.58	22.86 ± 0.87	27.3 ± 0.93
G20 + TA 4 mg	20	4	26.91 ± 7.59	20.18 ± 1.32	36.91 ± 3.69
G20 + TA 8 mg	20	8	36.26 ± 9.20	13.23 ± 0.88	50.78 ± 1.94
TA 1 mg	0	1	28.67 ± 13.31	88.23 ± 1.94	46.90 ± 1.55
TA 2 mg	0	2	55.18 ± 4.62	82.77 ± 1.81	61.57 ± 0.82
TA 4 mg	0	4	78.01 ± 13.33	58.51 ± 0.24	100.21 ± 4.83
TA 8 mg	0	8	94.18 ± 20.06	35.32 ± 0.27	114.91 ± 2.44

## Data Availability

Data are contained within the article.

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
