# Peer review of "In Situ Formation of Injectable Gelatin Methacryloyl (GelMA) Hydrogels for Effective Intraocular Delivery of Triamcinolone Acetonide"

_ijms, 2023, doi:10.3390/ijms24054957_

Round 1
Reviewer 1 Report
The manuscript entitled “In Situ Forming of Injectable Gelatin Methacryloyl (GelMA) 2 Hydrogels for Effective Intra-ocular Delivery of Triamcinolone 3 Acetonide” by Shen et al. is concerned with sustaining the drug effect of GelMA hydrogels coupled with Triamcinolone Acetonide 11 (TA) within the vitreous cavity. Although the authors exerted good efforts, I think the manuscript may be considered for publication in with few changes.
The Major points are:-
- The authors should incorporate an objective of the study in the introduction section of the manuscript
- In line 106, ‘Compact materials absorb less water and have a lower swelling rate’ the authors can be more specific on this, and please incorporate these details in the discussion section of the manuscript.
- Why did the authors treat HRPE cells specifically with 405nm light exposure? Please incorporate the details in the discussion section
- Deep grammatical revision should be carried out.
- Line 32, adjust the font size
Reviewer 2 Report
This manuscript reports in situ forming of injectable gelatin methacryloyl (GelMA) hydrogels for intra-ocular delivery of Triamcinolone Acetonide (TA). TA was released from GelMA-TA in a sustained manner in the vitreous. It also showed good biocompatibility, both in vitro and in vivo, after injection into the rabbits’ eyes for 2 months. Major revision is needed before being considered for acceptance.
1. How the hydrogels formed should be shown in a scheme in molecule level.
2. How can we see that TA particles were homogenized in each pore (Figure 1) as described in Line 88.
3. Is there any interaction between the hydrogels and TA.
4. More characterization of the hydrogels are needed.
5. The language need to be polished. Such as in Line 269-272: The pore size was dependent on the GelMA concentration, smaller pores could be observed with the increase of the GelMA concentration. The pore’s size decreased with an increase in GelMA concentration … The meaning is repeated.
Reviewer 3 Report
Interesting results with some new information from viewpoint of biological application. The paper focuses on injectable hydrogels based on Gelatin Methacryloyl containing triamcinolone acetonide as model drug for intraocular delivery.
Though the intention of the authors is highly commendable, there is a lot of problems particularly in the synthesis, physico-chemical characterization and ex-in vivo results presentation throughout the manuscript. Besides, there are many grammatical formulating mistakes throughout the manuscript and some results must be rewritten for a better understanding.
In view of the above comments, the whole manuscript should be properly written. Otherwise, the paper should be rejected, in the current state being below the standards of this journal.
There are many formulating errors through the entire manuscript, in particular the synthesis and characterization part must be reconsidered. Next, I exemplify just some of them.
1. In the abstract, please reconsider the statement “drug release from TA-hydrogels was more stable than drug release from TA suspensions”; what means “more stable”
The last phrase of the Abstract must be also reformulated.
2. Introduction section:
Page 2, line 56: “pH values” of whose?
Page 2, line 71: hydrogels have been used for bioadhesion??? Could be “as materials with bioadhesive properties”
Page 2, line 72: Photocrosslinking of GelMA obtained from natural hydrogel gelatin?
Page 2, line 87: to reduce side effects???
Page 5, line 147: G10 decreased the concentration in the hydrogel matrix?
Page 17, line 471: Please reconsider: „The hydrogels were immersed in culture medium at a ratio of 1 ml per 0.1 g gel surface area” –completely ambiguous
3. The authors must carefully verify and correct the Figures and their legend.
4. The entire discussion about the GelMA morphology must be rewritten.
5. The legend of Figure 1 is not in accordance with what the figure represents.
6. Figure 2: the authors state that swelling volume equilibrium of the hydrogels is reached after 8 hours. Figure 2C shows that this time is 12 hours. Why figure 2 B,C show the swelling data after 4 hours of immersion in PBS at 37 oC if the experiments were carried out for 8 hours (2B), respectively 12 hours (2D). In the legend of figure 2, panels G-H appear, which are not actually found in the figure.
7. The samples code must be consistent through the entire manuscript. They must define the term “prehydrogels” (in my opinion is not appropriate, better “precursor solutions”?); They must refer to the same term for GelMA (5, 10, 20%) or TA-loaded GelMA with different loading amounts of TA.
8. The drug release section is not clear discussed and in figure 3 must be represented also the effective experimental values (points), for a better understanding.
9. All ex-in vivo experiments are performed for GelMA hydrogels containing 2-8 mg TA. How the authors explain the use of 1 mg TA-loaded G10 hydrogels for the in vivo assays?
Round 2
Reviewer 3 Report
The authors have improved the manuscript by supplementary explanations; however, I have some comments as follows:
1. Page 1, lines 17-19: Please reconsider the phrase: “Rapid gel formation was observed after injection, and the in vitro release study confirmed that a slower and sustained-release kinetics in TA-hydrogels than TA suspensions”. The verb is missing.
2. Page 3, lines 113-115: Legend of Figure 1: The authors must modify the legend for panel (A) with “Schematic route of the synthesis of GelMA+TA hydrogel”
3. Page 5 line 156: Please be consistent with the annotation of the GeMA hydrogels: “On the other hand, the mean TA release from G10-1 …….observed for G20-1”. It must be G10+1mgTA and G20+1mgTA.
It is not appropriate the use of CONCENTRATION for hydrogels “Compared with G20, the concentration of the hydrogel matrix decreases in G10….” More probably they want to tell that” The loose structure of the G10 hydrogel favored a faster release and higher cumulative release of TA”
Page 3, lines 366: The phrase “In-vitro release level of TA from 1mg TA suspension WAS (this verb is missing) approximately the same as G10+4mgTA hydrogel….”
4. There are still a lot of grammatical errors. Please check it.
In conclusion, I recommend the publication after a minor revision.
